# An Optimal Network-Aware Scheduling Technique for Distributed Deep Learning in Distributed HPC Platforms

Sangkwon Lee [1], Syed Asif Raza Shah [2,3], Woojin Seok [4,*], Jeonghoon Moon [3], Kihyeon Kim [3] and Syed Hasnain Raza Shah [1]

[1] Science and Technology Information Science, University of Science and Technology, Daejeon 34113, Republic of Korea; sglee@kisti.re.kr (S.L.); hasnain@kisti.re.kr (S.H.R.S.)

[2] Department of Computer Science, CRAIB, Sukkur IBA University, Sukkur 65200, Pakistan; asif.shah@iba-suk.edu.pk

[3] KREONET, Korea Institute of Science and Technology Information, Daejeon 34141, Republic of Korea; jhmoon@kisti.re.kr (J.M.); kkh1258@kisti.re.kr (K.K.)

[4] Center for Quantum Communication, Korea Institute of Science and Technology Information, Daejeon 34141, Republic of Korea

[*] Correspondence: wjseok@kisti.re.kr

**Abstract:** Deep learning is a growing technique used to solve complex artificial intelligence (AI) problems. Large-scale deep learning has become a significant issue as a result of the expansion of datasets and the complexity of deep learning models. For training large-scale models, the cloud can be used as a distributed HPC (high-performance computing) tool with benefits in cost and flexibility. However, one of the major performance barriers in distributed deep learning in a distributed HPC environment is the network. The performance is often limited by heavy traffic like many stochastic gradient descent transfers for distributed communication. There are many network studies in distributed deep learning to solve these problems, but most research only focuses on improving communication performance and applying new methods or algorithms like overlapping parameter synchronization to minimize communication delay rather than considering the actual network. In this paper, we are focusing on the actual network, especially in a distributed HPC environment. In such an environment, if cluster nodes are assigned to different zones/regions which means a set of an appropriate number of distributed HPC nodes when performing distributed deep learning tasks, performance degradation due to network delay may occur. The proposed network optimization algorithm ensures that distributed work is placed in the same zone as much as possible to reduce network delay. Furthermore, scoring using network monitoring tools like loss, delay, and throughput is applied to select the optimal node within the zone. Our proposal has been validated on the Kubernetes platform, an open source orchestrator for the automatic management and deployment of micro-services. The performance of distributed deep learning is improved through the proposed scheduler.

**Keywords:** cloud computing; scheduling; container technology; distributed computing; network monitoring; deep learning; distributed HPC; AI



## 1. Introduction

Deep learning is used to solve complex AI problems. Nowadays, a variety of applications, such as computer vision, natural language processing, etc., are using deep learning [1]. However, the time required for learning has increased significantly as the dataset has become larger than before, and it requires more complex models. Recently, there has been a surge of research on distributed deep learning for these reasons. Distributed machine/deep learning is an effective way to shorten the increased time for learning. This method utilizes multiple nodes to speed up computing. The computation of training deep learning models has long outgrown the capabilities of a single high-end machine, leading

to distributed training being standard [2]. There are two major methods of distributed training. One is the server–client method using a parameter server [3,4], and the other is an all-reduce method using an MPI interface [5].

The architecture of the parameter server method consists of one or more parameter servers and a number of worker nodes. This method transmits parameter values obtained through calculations at multiple nodes to the parameter server to perform a gradient update. This process is shown in Figure 1; each connection from worker node 1 to M allocates training data through a minibatch process. The worker node works to find the gradient and sends the result to the parameter server. The parameter server multiplies the learning rate by the parameter value and adds it to the previous gradient value to create an updated gradient value. On the other hand, the all-reduce method is another way to mutually transfer parameter values between worker nodes without a server playing a central role. Figure 2 shows the all-reduce process on four worker nodes. As shown in the figure, in the first iteration, each worker n node sends an $N - 1$ segment to its next node. After the communication, the reduction operation is performed. The second iteration is the same; the only difference is it sends $N - 2$ segments. In the third iteration, after sending $N - 3$ segments to the next node, the n segment of each n node passes through all nodes and has a complete segment result. The method repeats this process to complete the remaining segments. The parameter server method is easy to implement, but it has problems such as the concentration of the network load on the server and low accuracy, whereas the all-reduce method is difficult to implement. Both methods have strengths and weaknesses.

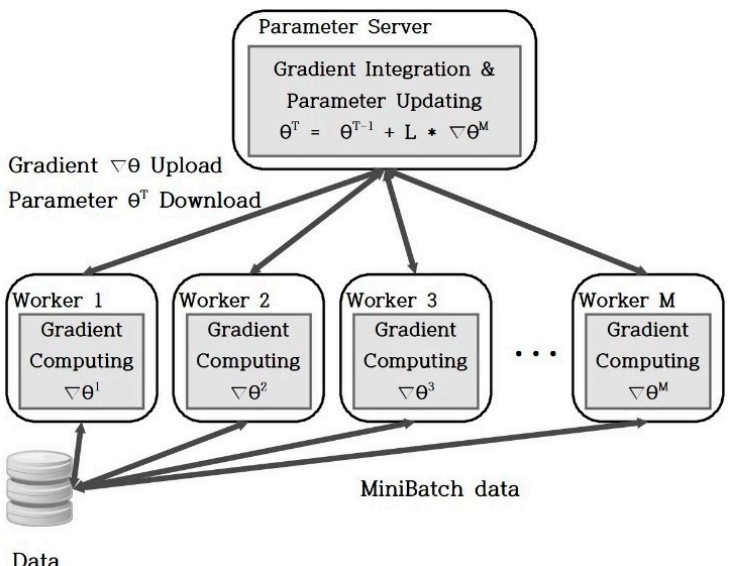

**Figure 1.** Architecture of parameter server framework [3].

Although distributed machine/deep learning uses multiple nodes, it usually has limited computing resources because it only utilizes the resources of one data center. As an alternative, distributed HPC (High Performance Computing) makes it possible to utilize multiple data centers. In this way, large-scale shared resources become available, and expansion of computing resources becomes easy as needed. In the past few years, a number of efforts have already been undertaken for distributed HPC platforms such as PRP (Pacific Research Platform), NRP (National Research Platform), and APRP (Asia Pacific Research Platform) [6]. These platforms are structured for the purpose of research data transfer between different countries, institutes, and universities. An example architecture of distributed HPC platform (APRP proposed project) is shown in Figure 3. The distributed HPC platform introduced above uses ScienceDMZ-based high-speed network. ScienceDMZ is a network infrastructure that serves as a highway for high-speed transmission of scientific

data [7]. A firewall-based network can cause the performance degradation in high-speed networks such as from 10 to 100 Gbps, so ScienceDMZ architecture can bypass such issues by implementing the simplest ACL (Access Control List) method on switches/routers. ScienceDMZ uses a DTN (Data Transfer Node) which is a specialized server for data transmission. DTN is equipped with a high-performance hardware resource like network card or CPU and is specially tuned for optimal performance. In APRP, distributed HPC is created by configuring a cluster with multiple DTNs. Using the multipurpose research platform, the researchers in Asian countries with low computing power can conduct research and utilize the platform for distributed machine/deep learning.

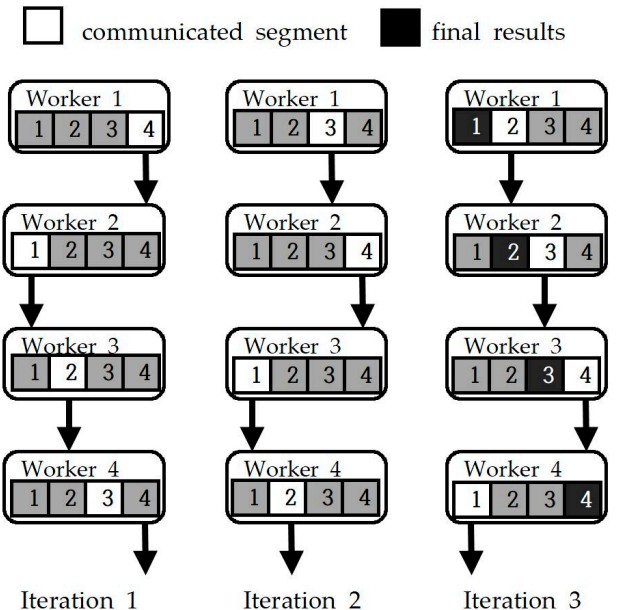

**Figure 2.** All-reduce algorithm for distributed deep learning [5].

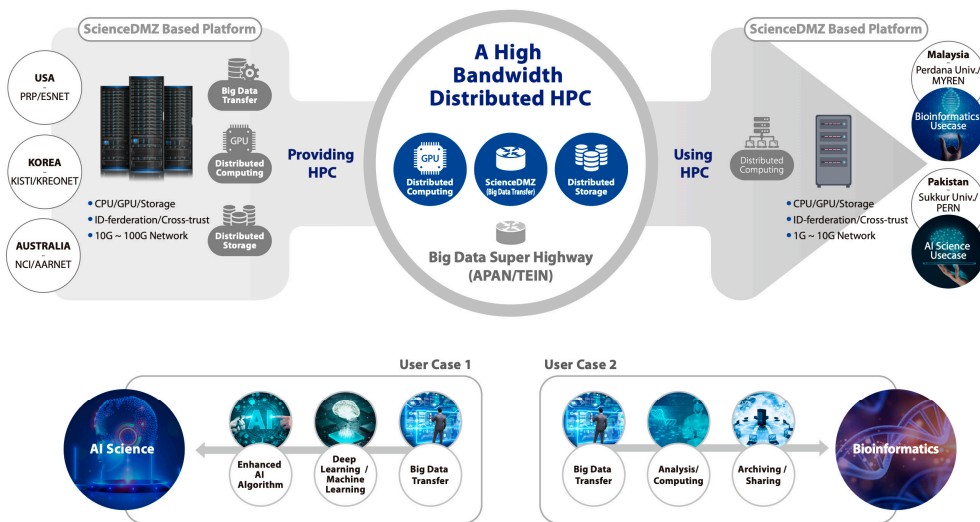

**Figure 3.** Architecture of distributed HPC platform [8].

Since these methods inevitably use multiple data centers and nodes, the network affects performance. In recent machine learning, as the size of data increases and complex algorithms are used, the number and size of parameters to be transmitted increase accordingly. As a result, performance benefits become limited by communication-heavy parameter synchronization step [9]. Furthermore, the computing power of computational units like GPUs grows faster than the growth of network performance. Now, network

performance is distributed through a training bottleneck [10]. In order to solve these problems, a lot of research has been conducted, such as a synchronization algorithm that can efficiently transmit parameter data and use as little bandwidth as possible. Through the approach of the analog encoding/decoding method, ML applications can be quickly performed at the wireless edge of low bandwidth [11]. To accelerate DNN training, BytePS proposes summation service and splits a DNN optimizer into gradient summation and parameter update [12]. Another paper proposes an ML job feature-based job scheduling for ML clusters running data and model parallelism ML jobs [13]. However, these studies are focused on communication methods and algorithms rather than considering actual networks. The algorithm we propose does not increase communication efficiency in distributed machine learning, but rather selects nodes with high network performance during machine learning works in the cloud. In this paper, we consider real networks, especially distributed HPC. When assigning a distributed machine/deep learning task in a cloud where nodes are deployed in multiple countries or regions, performance degradation occurs due to network delay if the nodes are deployed far apart.

To prevent this problem, this paper introduces a zone-based optimal network algorithm. The zones are assumed as regions or countries in this paper. If a distributed machine/deep learning task is placed, then a suitable zone with required CPU or GPU resources is selected, and all nodes are allocated the same zone as much as possible. In addition, when selecting as many nodes as needed in a zone, scores were given to network resources and elected according to priority. Network resources are measured using a network monitoring tool called perfSONAR [14], and the measured results can be gathered and viewed at a glance through a monitoring dashboard called MaDDash (Monitoring and Debugging Dashboard) [15]. The optimal node is selected by scoring with bandwidth, loss, and delay data between nodes provided in MaDDash. Through the proposed scheduler, the performance of distributed machine/deep learning can be improved.

This paper is organized as follows. Section 2 provides information about related work. Section 3 introduces the proposed solution, an optimal network algorithm. Section 4 sets the experiment testbed. We conduct experiments and present results. Finally, we conclude with Section 5.

## 2. Background and Related Works

### 2.1. Container Orchestration

Distributed HPC discussed in the paper utilizes container orchestration which is the process of managing and coordinating a large number of software containers across multiple servers to ensure that they are working together efficiently. A container is a lightweight and portable way to package and run software applications. Containers make it easier to move applications between different environments and ensure that they run consistently across different platforms. They enable developers to build and deploy software quickly and easily, without having to worry about dependencies or compatibility issues. However, managing a large number of containers can be challenging, especially when they are distributed across multiple servers. This is where container orchestration comes in. In simpler terms, container orchestration involves managing a large number of containers that are running on multiple servers, ensuring that they are all working together seamlessly and efficiently. Container orchestration platforms provide tools for automating tasks such as deployment and monitoring of containers, making it easier to manage large, complex container-based applications. These platforms use advanced algorithms to distribute container workloads across multiple servers to optimize resource utilization and ensure high availability of the application.

Some of the most popular container orchestration platforms include Kubernetes, Docker Swarm, and Apache Mesos. These platforms provide features such as container scheduling, automatic scaling, load balancing, service discovery, and fault tolerance. Overall, container orchestration simplifies the management of container-based application, making it easier for developers to focus on writing code and delivering value to their users, while ensuring that the applications are running smoothly and reliably.

Kubernetes is an open-source platform for managing and automating the deployment, scaling, and management of containerized applications. Kubernetes provides a platform for running containerized applications across a cluster of machines. It automates many of the tasks involved in managing containers, such as deployment, scaling, and monitoring. Some of the key features of Kubernetes include automatic scaling of containers to meet demand, load balancing of traffic between containers, automatic rolling updates and rollbacks of containers, and automatic placement of containers on available resources. Overall, Kubernetes makes it easier to manage and automate the deployment and management of containerized applications, allowing developers to focus on writing code and delivering value to their users, while ensuring that their applications are running smoothly and reliably.

In the paper, we utilize Kubeflow to perform distributed machine/deep learning. Kubeflow is a tool for building, training, and deploying machine learning models on Kubernetes, which is a platform for managing containerized applications. Machine learning involves training models on large datasets to make predictions or take actions based on new data. Kubeflow makes it easier to do this by providing a set of tools and services that enable developers and data scientists to build, train, and deploy machine learning models on Kubernetes. Kubeflow leverages the scalability and flexibility of Kubernetes to enable distributed training of machine learning models. This means that large models can be trained quickly and easily, without worrying about the underlying infrastructure.

### 2.2. Distributed HPC

The aforementioned distributed HPC is described in detail. HPC is a technology that configures multiple computing nodes into clusters to achieve high performance. The three factors that affect the performance of HPC are compute, network, and storage. To reduce the impact of the network, clusters are usually organized in one data center. However, some HPCs are configured with distributed computing nodes for other reasons, such as cost reasons or data transmission purposes. These HPCs are called distributed HPC. Projects such as PRP, NRP, and APRP are examples. These projects are originally focused on building an environment where scientific data can be transmitted and shared at high speed. Nodes in these clusters are physically separated but use high-bandwidth 10–100 Gbps networks and utilize network monitoring tools such as perfSONAR to compensate for network performance issues.

In PRP, Kubernetes, a container orchestration platform, is installed on the nodes of the cluster, and distributed HPC is used under the name of Nautilus [16]. Nautilus is a distributed HPC platform and collaboration between major research institutions in the United States aimed at providing researchers with access to advanced computational resources. Nautilus is designed to provide reliable access to large-scale data storage and processing capabilities, allowing them to analyze complex datasets and run simulations and other computational models. The platform includes a variety of computing resources, including high-speed networking, large-scale data storage, and high-performance computing clusters. One of the key features of Nautilus is its scalability, which allow researchers to easily scale their computational workloads up or down as needed. The platform is an important resource for researchers in a wide range of fields, including physics, biology, chemistry, and engineering, providing them with the computational capabilities.

Like this project, container-based cloud technologies such as Docker and Kubernetes are being used a lot these days, so they are becoming important. Recently, cloud orchestration platforms such as Kubernetes have been configured and used so that these research data can be used for computational tasks such as AI. Nautilus used by PRP is one such example. In Korea, a platform called R&E (Research and Education) Together was developed by benchmarking PRP and Nautilus. The R&E Together infrastructure consists of a big data transmission system and an AI orchestration system. Figure 4 shows the infrastructure of R&E Together. The DTNs used in the infrastructure were built using the servers of each research institute and university in Korea, and they are used to provide an AI research

environment. This distributed HPC project was expanded and developed into APRP, a pan-national distributed HPC connecting Asian countries such as Pakistan and Malaysia.

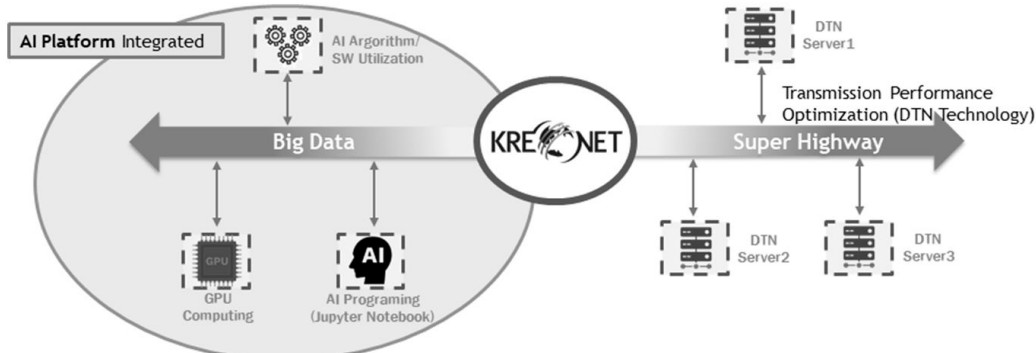

**Figure 4.** R&E Together research infrastructure diagram [7].

### 2.3. Distributed Machine/Deep Learning

Machine learning can be performed using distributed HPC to reduce training time by increasing computing performance. In this case, the process of exchanging parameter values between nodes and communication is essential, so the network can affect the learning performance. For this reason, there are many studies to improve network efficiency. A brief description of related studies is as follows. As distributed machine/deep learning is utilized, the resources required that must be considered include CPU, GPU, Disk I/O, and Network I/O. However, existing Deep Learning (DL) schedulers focus on only GPU allocation.

The following are related studies. Optimus [17] is an efficient dynamic resource scheduler for deep learning clusters. It predicts model convergence and estimates training performance for efficient scheduling. In distributed machine/deep learning, when a server is overloaded with traffic from multiple nodes, it shows the result of performance degradation. So, this scheduler predicts the appropriate number of servers according to the ratio of worker nodes so that traffic can be distributed. Tiresias [18] is a GPU cluster manager for distributed deep learning. The key idea is utilizing a two-dimensional metrics framework that aims to minimize job completion time (JCT) when a DL job's execution time is unpredictable.

However, considering only GPU resources has the following limitations. The problem is that it can take a long time to complete because it has to wait when certain resources are shortage. Muri [19], a multi-resource cluster interleaving of DL workloads, can help this problem. Scheduling based on Blossom algorithm for multi-resource multi-job packing reduces JCT by maximizing interleaving efficiency. However, multi-resource interleaving has different effects on different jobs, which could cause a fairness problem.

But there is no study on how much bottleneck exists [20]. This study tests a real network. The scale factor was calculated by taking the actual network bandwidth Tn when n node was added and the network bandwidth NT when n nodes are used. This value is close to 100%, so if the actual bandwidth of the node is high, all performance can be brought without considering the network. On the other hand, in a low-speed network of 10 Gbps or less, the network must be optimized to achieve full performance.

Large data centers use high-speed networks such as 40/100 Gbps Ethernet to alleviate the communication delay, but many researchers and small data centers are still using consumer-level GPUs connected by low-bandwidth networks such as 1 Gbps Ethernet [21]. These factors make distributed machine/deep learning networks running on distributed HPC more important.

## 3. Proposed Solution

Our proposed solution aims to optimize the performance of distributed deep learning workflows in a distributed HPC environment by considering the actual network. Our pro-

posed algorithm focuses on selecting the appropriate nodes in Kubernetes based on their availability in different zones and their available resources such as CPUs, GPUs, and memory. Our proposed optimized scheduling mechanism will ensure that the distributed work is placed in the same zone as much as possible to minimize network delay. Additionally, it uses network monitoring tools to score and select the optimal node within the zone.

The proposed solution has been validated on the Kubernetes platform, an opensource orchestrator for the automatic management and deployment of microservices. The proposed optimal scheduler has improved the performance of distributed deep learning workflows in the distributed HPC environment. In this section, we discuss the proposed network monitoring architecture for distributed HPC platform and an optimal networkaware scheduling algorithm.

### 3.1. Monitoring Architecture for Distributed HPC Platform

As illustrated in Figure 5, let us suppose that there is a distributed high-performance computing (HPC) platform operating across multiple countries including the United States, South Korea, Pakistan, Australia, and Malaysia. This platform consists of worker nodes that are deployed in all of these countries, while the master node is located in South Korea and connected to a high-speed network. For such types of distributed HPC scenarios, we proposed a comprehensive network monitoring architecture that is helpful for scheduling the distributed deep learning pods on Kubernetes cluster in an efficient way. In order to achieve the high performance across the distributed HPC platform, all of the worker nodes should be configured as specialized Data Transfer Nodes (DTNs) along with the ScienceDMZ architecture. To further optimize the network's performance, we have developed a perfSONAR network monitoring system for our platform. Specifically, we have deployed the MaDDash monitoring system and installed perfSONAR testpoints on all worker nodes. This enabled us to effectively monitor the network's performance in real-time and ensure that the platform is operating at optimal levels.

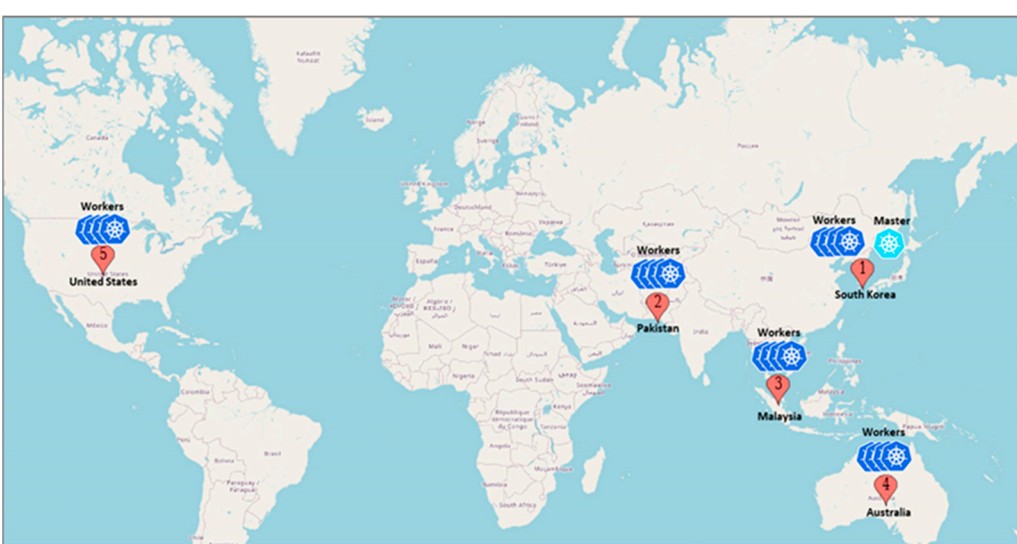

**Figure 5.** An example of distributed HPC platform among countries.

Overall, this distributed HPC platform represents a complex and highly advanced technological infrastructure, but with our proposed network monitoring system in place, we can ensure that it operates efficiently and is helpful to the network-aware scheduling algorithm.

The overall network monitoring architecture for distributed HPC platform can be represented as in Figure 6, where a zone has a number of Kubernetes worker nodes which received tasks from the Kubernetes master node and executed them in pods. All Kubernetes worker nodes have perfSONAR testpoints containers running. The perfSONAR testpoints container includes measurement tools such as iperf, ping, nuttcp, and traceroute and runs

tests according to the scheduler daemon. Network performance data obtained by test is sent to the MaDDash server. The MaDDash server archives the data, and it can be checked at a glance as shown in Figure 7. These graph values support an interface so that they can also be obtained through REST APIs. Our proposed network-aware scheduling algorithm uses REST APIs to get network performance data from the MaDDash server. The scheduler selects the optimal node(s) through scoring based on the network performance result and delivers the selected node list to the Kubernetes master server (described in Section 3.2). Based on this, the Kubernetes Master can perform distribute machine learning tasks by selecting nodes with low network latency/loss and high available bandwidth.

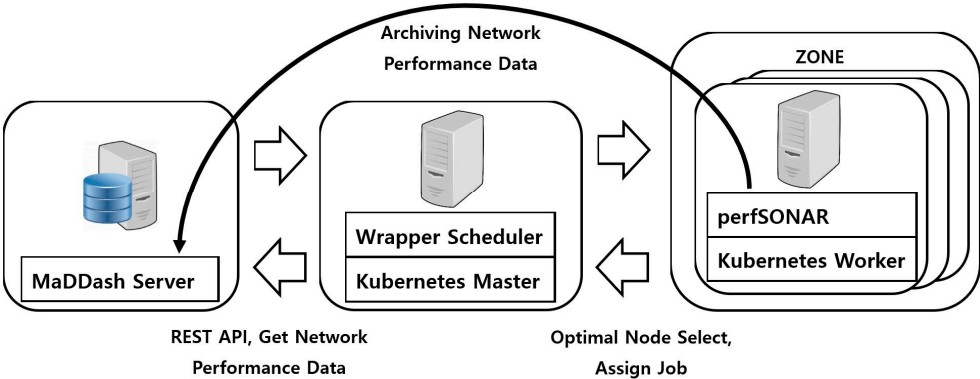

**Figure 6.** Monitoring Architecture for Distributed HPC Platform.

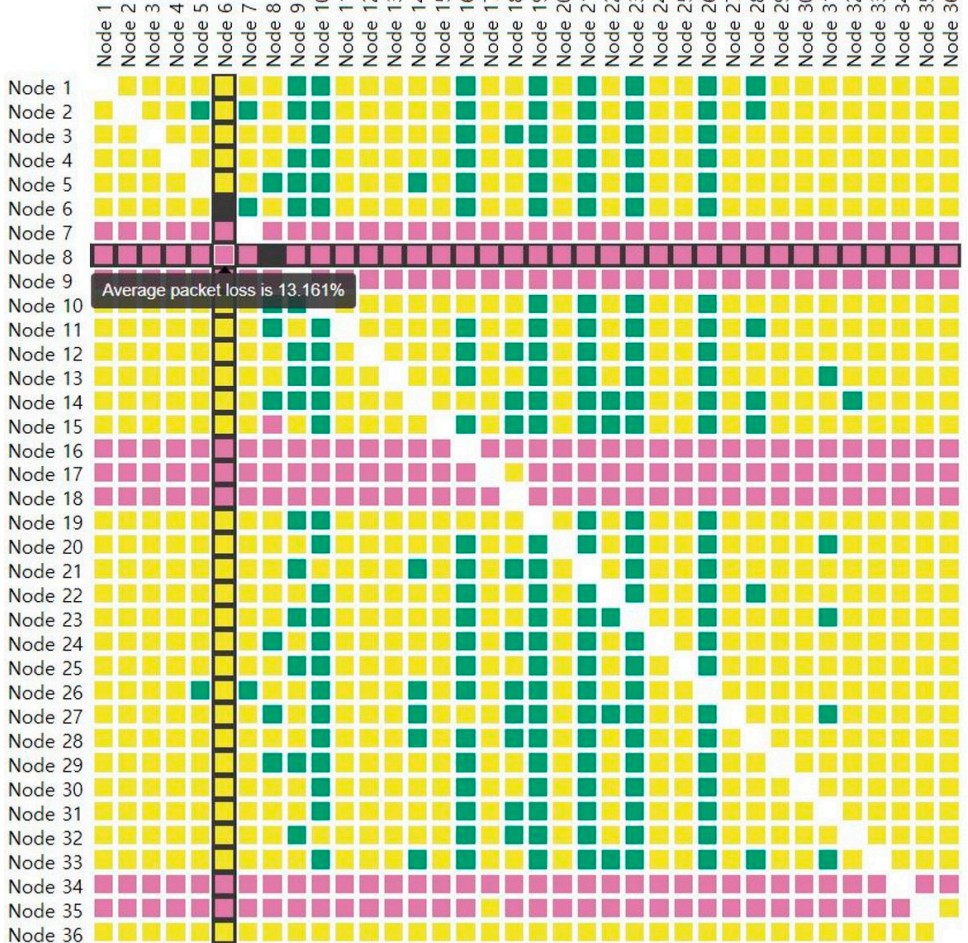

**Figure 7.** An Example of DTN Nodes Report on MaDDash. (Green: loss rate is ≤0.001%; yellow: loss rate is >0.001%; pink: loss rate is ≥0.1%).

### 3.2. An Optimal Network-Aware Scheduling Technique

In this section, we explain optimal network aware scheduling technique. As mentioned in the previous chapter, we utilize Kubernetes and Kubeflow platform to perform distributed machine/deep learning. If distributed machine/deep learning is performed on Kubernetes, a suitable node is selected through the node selection process. Node selection in Kubernetes is the process of determining which worker node in a cluster should run a particular pod, which is a group of one or more containers that are deployed together. When a pod is created, Kubernetes uses a process called scheduling to determine which node in the cluster should run it. Scheduling involves selecting a node that has enough resources such as CPUs, GPUs, and memory available to run the pod. The scheduler also uses variety of factors to select the best node for a pod—for example, resource limits, node affinity, taints, tolerations, etc. However, the Kubernetes default scheduler selects nodes without considering the proper network parameters, which affects distributed machine/deep learning performance.

In this paper, we proposed a novel scheduling technique to consider the network performance between nodes and utilized the distributed HPC platform in an optimal way. Our proposed scheduling technique allows users to customize the way jobs are scheduled on a computing cluster. In a computing environment, a job scheduler manages the allocation of resources to user jobs. By using our proposed wrapper scheduler, the users can define their own scheduling policies that are applied on top of the existing job scheduler. This allows users to prioritize or balance jobs based on their specific needs.

A wrapper scheduler has a name for each data center identified as a zone. Since each zone is a data center divided by country or region, there may be network delay between zones. Therefore, when selecting nodes, the scheduler selects only nodes in the same zone if possible. The scheduler selects a zone after first checking that the zone has enough resources to run the job. When a zone is selected, the optimal node in the zone is selected. When selecting the optimal node, it takes a lot of time to check the network values of all nodes in the cloud. In the algorithm we proposed for this problem, the amount of calculation is reduced by using a scoring process of only group data divided into zones. Optimal node selection is done through scoring with network performance. Network performance scoring is based on data using perfSONAR, a network performance measurement tool, and MaDDash, a network monitoring tool.

$$\text{SelectNode} = \text{Top-ranked}$$

$$\text{Top-ranked} = \text{Highest score of } f(t) \text{ in (Equation(4))}$$

The data used are bandwidth, loss, and delay. These values are collected through network performance monitoring in MaDDash. Equations (1)–(4) define calculating scoring as follow. Scores are calculated using an approximate ratio of 20:40:40. These values are set so that the total sum becomes about 100 as a standard. Each ratio can be modified through the scheduler as needed. Delay and loss are calculated as negative values because the network performance deteriorates as the value increases. Each value can be adjusted to a suitable value by adding a weight value separately. Finally, each calculated result is added together to obtain a final score. In Equation (1), $t_\alpha$, $v_\alpha$, and $w_\alpha$ represent loss score, loss value, and loss weight, respectively. In Equation (2), $t_\beta$, $v_\beta$, and $w_\beta$ represent delay score, delay value, and delay weight. In Equation (3), $t_\gamma$, $v_\gamma$, and $w_\gamma$ represent bandwidth score, bandwidth value, and bandwidth weight. In the case of loss and delay, the higher the value was, the lower the network performance was, so the value was minus when calculating the score. On the contrary, the higher the network bandwidth was, the higher the network performance was, so the value was multiplied. In Equation (4), $f(t)$ represent total scores.

The following is the score of an equation:

$$t_\alpha = 20 - v_\alpha \cdot w_\alpha \tag{1}$$

$$t_\beta = 40 - v_\beta \cdot w_\beta \tag{2}$$

$$t_\gamma = 40 \cdot v_\gamma \cdot w_\gamma \tag{3}$$

$$f(t) = \sum_{i=\alpha}^{\gamma} t_i \tag{4}$$

As illustrated in Figure 8, our proposed network-aware scheduler is selecting zones and servers based on resource availability and network parameters as follows:

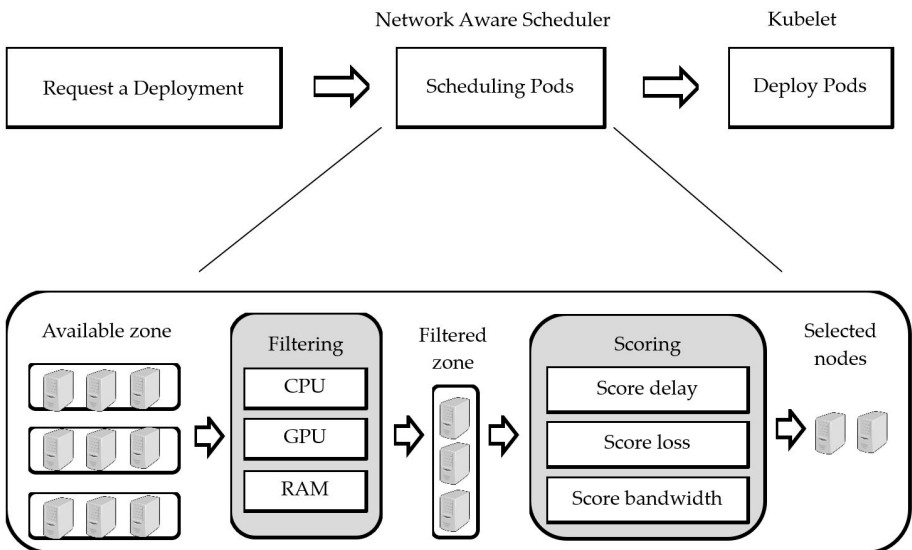

**Figure 8.** Network-aware Scheduling Technique for Node Selection Process.

Gather information about the available zones, servers, and their resources.

1. Determine the requirements of the task in terms of resources such as CPUs, GPUs, memory, etc.
2. Filter the zones based on their proximity to the intended users and select the nearest zone.
3. Filter the servers within the selected zone based on the required resources and their availability.
4. Rank the remaining servers based on the network parameters such as delay, loss, and bandwidth.
5. Select the top-ranked servers and assign the task to them.

Here is a more detailed explanation of each step:

Step 1: Gather information about the available zones, servers, and their resources.

You will need to have access to data about the available zones, servers, and their resources. This information can be obtained from a cloud provider or a data center. You will need to know the number of servers, the type of CPUs, GPUs, and memory available on each server and the network parameters for each server.

Step 2: Determine the requirements of the task in terms of resources such as CPUs, GPUs, memory, etc.

You need to define the requirements of the task in terms of the resources it needs to execute. This information can be provided by the user or can be automatically determined by the algorithm.

Step 3: Filter the zones based on their proximity to the intended users and select the nearest zone.

You need to filter the available zones based on their proximity to the intended users. This will help minimize the latency and improve the user experience. Once you have filtered the zones, you can select the nearest zone to the user.

Step 4: Filter the servers within the selected zone based on the required resources and their availability.

Once you have selected the zone, you need to filter the available servers based on their availability and the required resources. This will help ensure that the task is executed on a server that has the required resources and is available to execute the task.

Step 5: Rank the remaining servers based on the network parameters such as delay, loss, and bandwidth.

Once you have filtered the servers, you need to rank the remaining servers based on the network parameters such as delay, loss, and bandwidth. This will help ensure that the task is executed on a server with the best network performance.

Step 6: Select the top-ranked servers and assign the task to them.

Once you have ranked the servers, you can select the top-ranked servers and assign the task to them. This will help ensure that the task is executed on the best available server with the required resources and the best network performance.

The algorithm of step 5, which explains the core of the network scheduler, is the pseudo code below. This code is an algorithm that selects the optimal node according to the network performance score of each node. A step-by-step description of the node selection process is as follows.

1. It receives the node list of the zone and the required number of nodes as input values.
2. When checking the network performance between nodes, it is not necessary to check the same nodes, so the first node in the node list is included in the exclusion list.
3. Check the network performance between all nodes in the zone. At this time, both sides are checked; high values are used for loss and delay; low values are used for bandwidth; and scores are obtained using the formula mentioned above.
4. Scores are summed to calculate a total score. At this time, if there are previous combined scores, they are added and stored.
5. The total score is saved again to the previous total score.
6. The node with the highest total score is selected, and that node is added to the exclusion node list.
7. After that, the number of nodes to be selected is additionally selected through the process of 4–6 again.

---

**Pseudo code:** Scheduling for Selecting Nodes

---

```
scheduleSelectNode(Nodelist[], NeedNodeNumber)
#Get Available NodeList[] and NeedNodeNumber

for j in range NeedNodeNumber
    for i in range nodeinzone
        if NodeList[i] not in ExclusionList
```

$\quad\quad\quad\quad$ **if** $v_\alpha \geq v'_\alpha$ and $20 - v_\alpha * w_\alpha > 0$

$\quad\quad\quad\quad\quad\quad t_\alpha = 20 - v_\alpha * w_\alpha$

$\quad\quad\quad\quad$ **else if** $v_\alpha < v'_\alpha$ and $20 - v'_\alpha * w_\alpha > 0$

$\quad\quad\quad\quad\quad\quad t_\alpha = 20 - v'_\alpha * w_\alpha$

$\quad\quad\quad\quad$ **else**

$\quad\quad\quad\quad\quad\quad t_\alpha = 0$

$\quad\quad\quad\quad$ **if** $v_\beta \geq v'_\beta$ and $40 - v_\beta * w_\beta > 0$

$\quad\quad\quad\quad\quad\quad t_\beta = 40 - v_\beta * w_\beta$

$\quad\quad\quad\quad$ **else if** $v_\beta < v'_\beta$ and $40 - v'_\beta * w_\beta > 0$

$\quad\quad\quad\quad\quad\quad t_\beta = 40 - v'_\beta * w_\beta$

$\quad\quad\quad\quad$ **else**

$\quad\quad\quad\quad\quad\quad t_\beta = 0$

$\quad\quad\quad\quad$ **if** $v_\gamma \geq v'_\gamma$

$\quad\quad\quad\quad\quad\quad t_\gamma = 40 * v'_\gamma * w_\gamma$

---

> **else**
> > $t_\gamma$ = 40 * $v_\gamma$ * $w_\gamma$
> **if** NodeList[i] in pretotal[]
> > total[i] = pretotal[i] + $t_\alpha$ + $t_\beta$ + $t_\gamma$
> **else**
> > total[i] = $t_\alpha$ + $t_\beta$ + $t_\gamma$
> pretotal[] = total[]
> Node = max(total[])
> \# Selected nodes are excluded from scoring for the next selection
> selectlist append Node
> ExclusionList append Node

**Return** SelectList[]

## 4. Experiments and Results

This section outlines the experimental design and results obtained in a network-aware scheduler testbed. The cluster was configured with multiple servers to utilize Kubernetes technology.

### 4.1. Experimental Testbed

Figure 9 shows the network-aware scheduler testbed. The testbed consisted of a total of 38 servers and 2 storage arrays. Of these servers, one was designated as the Kubernetes Master node, while another server, equipped with two storage arrays, was designated as the Kubernetes Storage node. The remaining 36 servers were designated as Kubernetes Worker nodes, which were divided into 4 groups, each identified by the names zone A, zone B, zone C, and zone D, to add the concept of a zone.

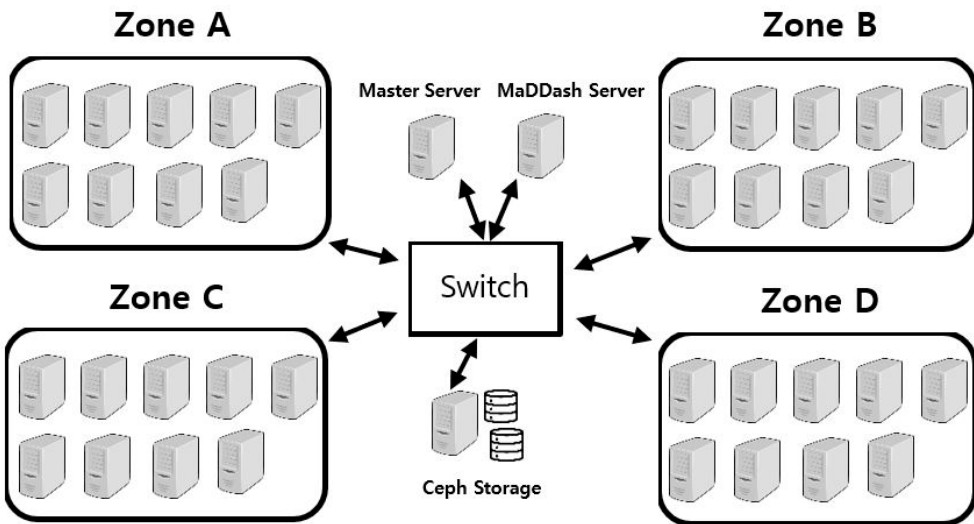

**Figure 9.** Network-aware Scheduler Testbed.

To enable monitoring of the network performance of each node, a perfSONAR container was created and executed through Kubernetes daemonset. The MaDDash server collected the network performance data through the REST API and presented it in the format shown in Figure 7.

The network-aware scheduler ran on the Kubernetes master server, taking a list of filtered nodes and the number of nodes required to run distributed machine/deep learning as input. The scheduler then scored nodes using network performance data retrieved from the MaDDash server through the REST API. The list of selected nodes was subsequently sent to the kubelet to create pods.

In summary, the experimental setup comprised a cluster of 38 servers and 2 storage arrays configured to utilize Kubernetes technology. A perfSONAR container was created to monitor network performance, and the MaDDash server collected the data through the REST API. The network-aware scheduler scored nodes based on this data and selected the nodes required to run distributed machine/deep learning.

In the test, notMNIST and CIFAR-10 model images, which are open used as examples, were used. The notMNIST dataset consisted of handwritten characters from A to J. This dataset is often used as a benchmark for character recognition algorithms. The notMNIST dataset contains 500,000 images, each of which is 28 × 28 pixels. The CIFAR-10 is a dataset of labeled images used for object recognition research. The CIFAR-10 dataset contains 60,000 images, each with a resolution of 32 × 32 pixels, and each image is labeled with one of ten classes: airplane, automobile, bird, cat, deer, dog, frog, horse, ship, or truck. The test took these two dataset and measured time while varying network performance such as bandwidth, delay, and loss.

Our proposed scheduler improves efficiency by excluding nodes with network performance that adversely affect distributed machine/deep learning. To this end, the scoring of node network performance was used in the scheduler, and it was tested that nodes with low network performance were actually excluded. The process of scoring all the network performance between each node takes a lot of time because it needs to be calculated as much as $N \times N$ when N is the number of nodes. We improved and applied this to $N + (N - 1) + (N - 2) + \ldots + (N - i + 1)$ when i is number of Nodes required for distributed machine/deep learning to shorten time.

### 4.2. Test Case I: Nodes Selection Using Scheduler

The selection test was conducted as follows. In Zone D, which has nodes 28 to 36, the TC (Traffic Control) tool was used for nodes 33, 34, and 35 to set delay of 10 ms, loss of 10%, and bandwidth of 10 Mbps, respectively. TC is a network quality-of-service tool that is included in the Linux kernel. It allows network administrators to control and prioritize network traffic on a Linux system. Zone D consists of 9 nodes, and if we enter the scheduler to select 6 nodes, nodes 33, 34, and 35, which are set to have the lowest network performance, should be excluded. This test proceeds as shown in Figure 10 below. In the process of checking the first network performance N times, node 32 with the highest score is selected. In the second network performance check, node 32 is excluded, so it is checked $(N - 1)$ times. In this case, the score is large because it adds the score of the first performance check. Node 31 with the highest score is selected. It proceeds according to this process, and at the end, $(N - i + 1)$ network checks are performed. Finally, Node 29 was selected, and nodes 33, 34, and 35, which had the lowest network performance, were excluded from node selection as expected. So, the final selected nodes are 28, 29, 30, 31, 32, and 36, which add up to 6 nodes.

The test is conducted using the aforementioned distributed machine/deep learning image and the proposed scheduler. Zones are divided into cities or countries. It is assumed that there is a problem in network performance between different zones, so that distributed machine/deep learning is executed mainly in the same zone. In this environment, our scheduler's goal is selecting nodes with optimal network performance within the zone and excluding nodes with network problems. The zone to be tested consists of 9 nodes. Network problems were created using TC tool on 3 nodes in the zone. We conducted the test through scenarios assuming problems such as low bandwidth, high loss, and high delay for 3 nodes. It then runs distributed machine/deep learning jobs that utilize 6 of the 9 nodes in the same zone.

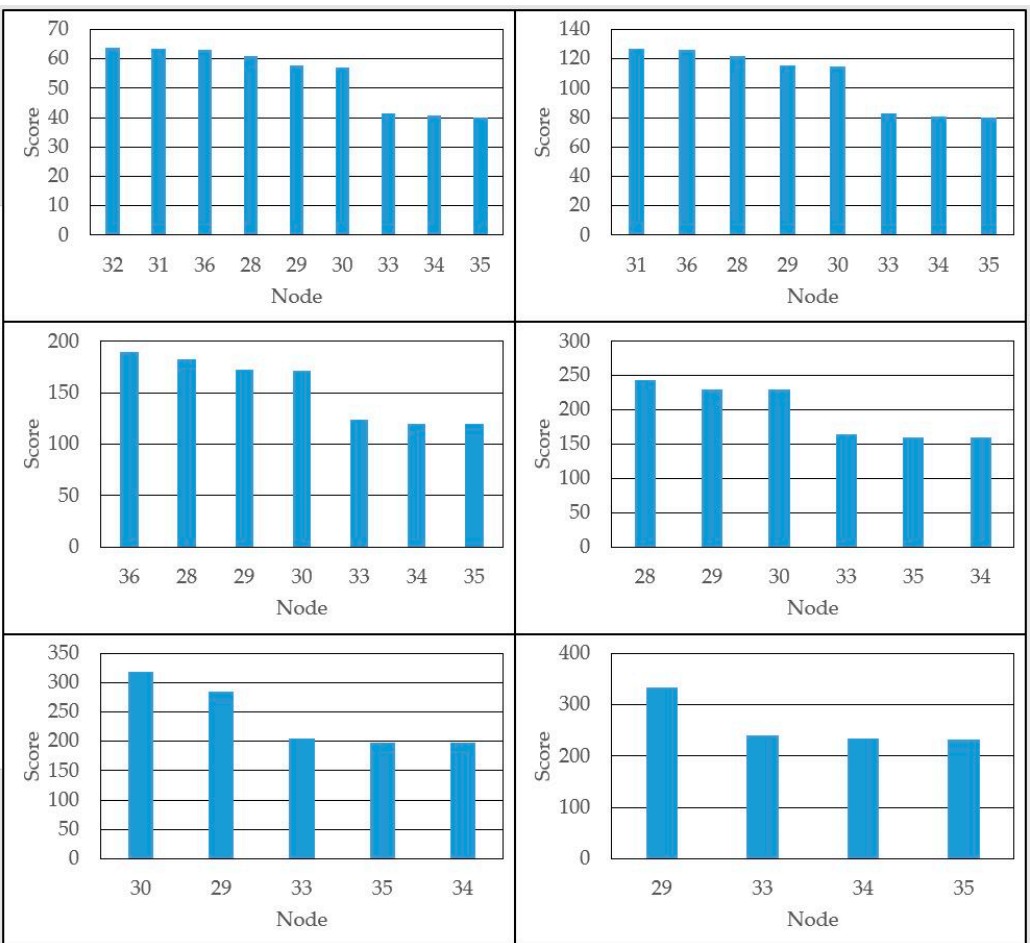

**Figure 10.** Selection of nodes through scoring.

### 4.3. Test Case II: Low Bandwidth Scenario

In the first scenario, low bandwidth is set on three nodes, and learning time is checked after running distributed machine/deep learning. The test results are shown in Figure 11 below. Distributed machine/deep learning selected six out of nine nodes in the zone to run. Since the default scheduler in Kubernetes does not consider network performance, only factors such as CPU, memory, or image locality, there are too many cases, so this test only considers the case of selecting all three worst nodes which adjusted network performance. We call the case where the default scheduler selects all three worst nodes as the worst case. Conversely, when the proposed scheduler was used, it was marked as proposed. Change the network bandwidth of three nodes from 40 Mbps to 90 Mbps and test. All other nodes operated at 100 Mbps performance on average. The proposed scheduler in the experimental results shows constant performance regardless of changes in the three nodes because it excluded three nodes with network performance problems. On the other hand, since the worst case of the default scheduler assumes that all three nodes with problems in the network are selected, different results are shown when the three-node network changes. In particular, notMNIST shows no change in bandwidth, but cifar10 shows the worst performance of 1041 s in 40 Mbps bandwidth. The reason for this tendency is that the image size of the CIFAR-10 dataset is $32 \times 32$, which is larger than that of notMNIST, which is $28 \times 28$, so it is more affected by network performance.

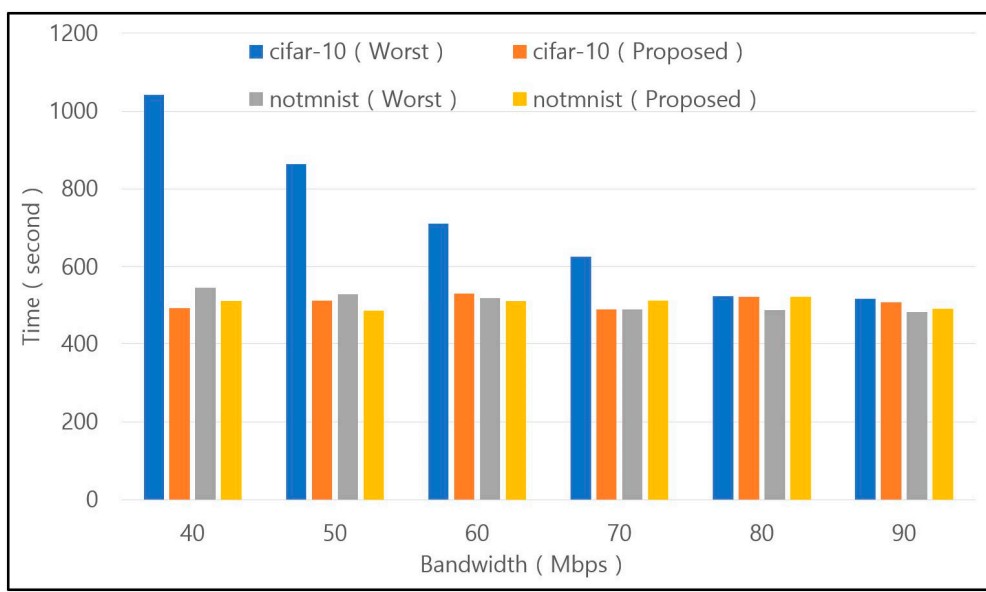

**Figure 11.** Performance Test with Low Bandwidth.

*4.4. Test Case III: Higher Loss Scenario*

In the second scenario, high loss is set on three nodes. The test results are shown in Figure 12 below. In the loss test, proceed while adjusting from 5 to 50. Similar to the bandwidth test, in the worst case of default scheduler, the learning time of cifar10, which is sensitive to network performance, increased first, and as the loss more increased, the learning time of notMNIST also increased. The proposed scheduler succeeded in excluding these nodes and showed consistent performance.

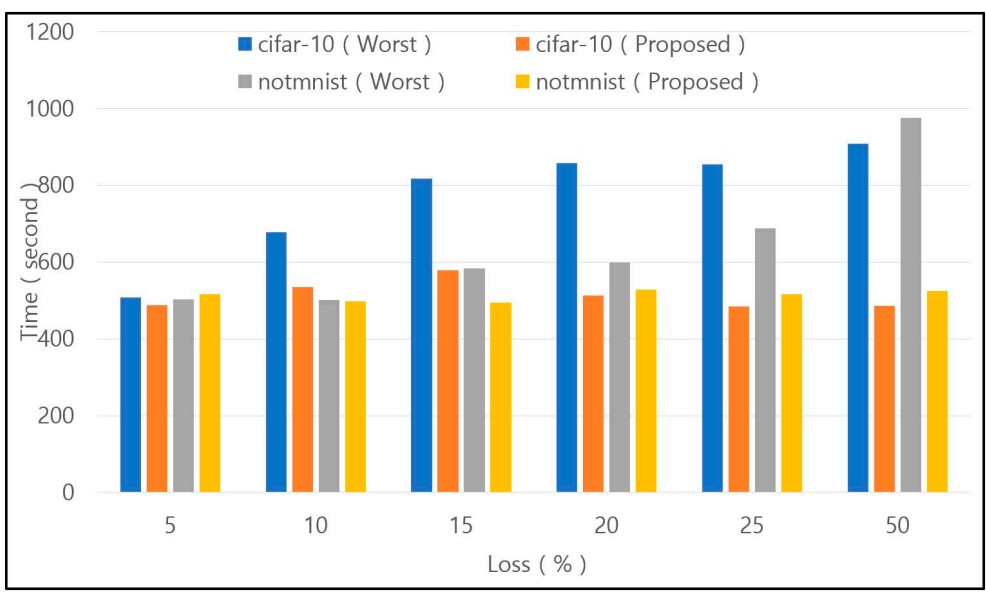

**Figure 12.** Performance Test with High Loss.

*4.5. Test Case IV: Higher Delay Scenario*

Finally, the last scenario is high delay on three nodes. The test results are shown in Figure 13 below. The delay was tested by adjusting from 50 to 500 ms. In the worst case in the default scheduler, the learning time of cifar10 and notMNIST tends to increase together. But this test has less impact on performance than loss. So even at 500 ms, the worst case does not exceed 800 s. On the other hand, the proposed scheduler succeeds in excluding three nodes with high delay and shows a constant learning time.

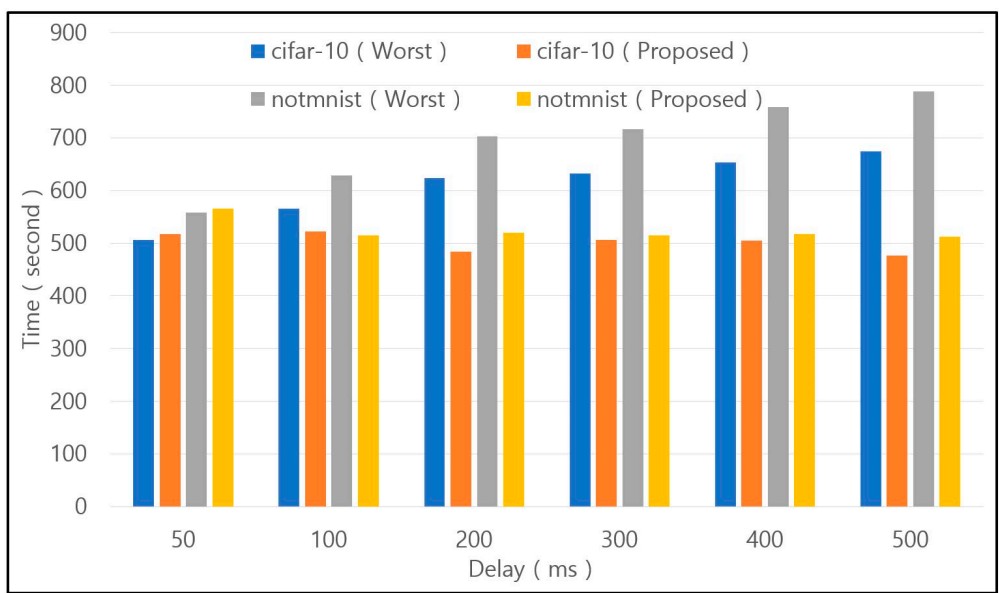

**Figure 13.** Performance Test with High Delay.

*4.6. Performance Improvement*

Based on the results tested through the scenarios above, we analyze the performance gains. The result is shown in Figure 14 below. In the bandwidth graph, the average learning speed is calculated using the time taken for learning, and the learning speed of the worst case of the default scheduler is set as 100% and compared with the performance of the proposed scheduler. As a result, the proposed scheduler improved by 40% in cifar10 and by around 1% in the case of notMNIST. Like the bandwidth graph, loss and delay are also based on the learning rate of the worst case of the default scheduler as 100%. In case of loss, the proposed scheduler shows a performance improvement of 50% for cifar10 and 25% for notMNIST. Finally, the proposed scheduler in delay shows a performance improvement of 21% for cifar10 and 32% for notMNIST. As these results show, if the proposed scheduler is used, it is possible to perform distributed machine/deep learning by excluding nodes with poor network performance in the zone, and thus, performance can be expected to improve.

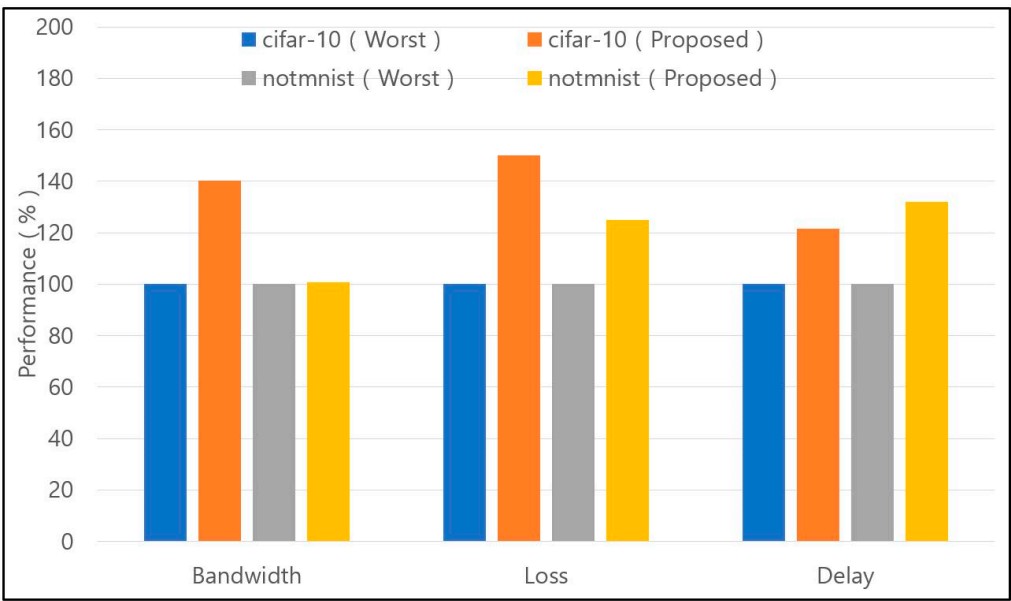

**Figure 14.** Performance Improvement.

## 5. Conclusions

The proposed scheduler selects the optimal node in consideration of network performance in node selection for distributed machine/deep learning execution in a cloud environment such as Kubernetes. Recently, a lot of data centers are configured with container-based clouds likes Kubernetes. Kubernetes supports scaling across multiple regions. We classified these physically separate data centers as zones. For example, the data center in Korea and the data center in Malaysia are far apart. Network performance between them is low, making it difficult to run tasks where network performance is critical, such as distributed machine/deep learning. The proposed scheduler guarantees that all pods run in one zone as much as possible after filtering required resources such as CPU, GPU, and memory when running distributed machine/deep learning jobs. What is especially important here is that even within the same zone, the optimal nodes must be selected because the network bandwidth, loss, and delay are different. The proposed scheduler uses data from network monitoring tool called MaDDash, which is used in high-speed research networks such as PRP, APRP, and NRP, to find the optimal node through scoring. Through this process, nodes with problems in network performance are excluded to show better performance when running distributed machine/deep learning.

**Author Contributions:** Conceptualization, S.A.R.S.; methodology, S.A.R.S. and S.L.; software, S.A.R.S. and S.L.; validation, S.A.R.S., S.L. and K.K.; formal analysis, S.A.R.S., S.L. and S.H.R.S.; resources, W.S. and J.M.; writing—original draft preparation, S.A.R.S. and S.L.; writing—review and editing, all authors; supervision, W.S. and S.A.R.S.; funding acquisition, W.S. and J.M. All authors have read and agreed to the published version of the manuscript.

**Funding:** This research was supported by Quantum Cryptography Communication based Networking Infrastructure Project funded by Korea Institute of Science and Technology Information (K-23-L04-C02-S01).

**Acknowledgments:** The authors would like to extend our sincere thanks to Global Science Experimental Data Hub Center at Korea Institute of Science Technology Information for supporting our research and providing the experimental environment.

**Conflicts of Interest:** The authors declare no conflict of interest.

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
