# Peer review of "An Optimal Network-Aware Scheduling Technique for Distributed Deep Learning in Distributed HPC Platforms"

_electronics, doi:10.3390/electronics12143021_

Round 1

Reviewer 1 Report

In this paper, the authors proposed a method to schedule distributed deep learning in distributed HPC platforms. The following lists some comments. The introduction of optimal scheduling need be strengthened, especially in its mathematical forms (optimization). The second is the proposed heuristic method to solve the NP problem. The method need be introduced further to show the main approximation. Third is about the comparison of different methods or strategies. In the Figures 10-13they are not very clear. And the other methods need be included, such as the ones without the network-aware method.

The language can be improved.

Reviewer 2 Report

In this manuscript, the authors proposed a network optimization algorithm for distributed deep learning in distributed HPC platforms. The experimental results demonstrate the effectiveness of the method. This research has certain application value. There are some issues that need to be addressed before it is ready to be published.

1. The introduction should provide a more comprehensive analysis of existing methods.

2. The basis for the selection of parameters in the experimental section should be described.

3. The quality of figure 10 needs to be improved.

Reviewer 3 Report

The idea of the article is that in multiple data centers, scheduler picks an "optimal" nodes considering the dynamic network characteristics.

Overall the approach is interesting, Kubernetes is also a hot topic. However, the article is not yet ready for publication. I propose to ask for a major revision improving presentation and motivation.

Pro/Cons:

+ Being able to run Kubernetes on 5 sites across countries is generally interesting

+ Using probing to identify suitable nodes that can collaborate on a task

- Motivation must be improved (see below)

- Presentation: English should be improved, figures are blurry, abstract is lacking details

- Scheduling algorithm doesn't do necessary what is promised (see below)

== Motivation ==

The article motivation for using multiple sites and for the use case are not convincing. Data centers have 100's to 1000's of GPUs. Why would one need DHPC then?

Using reduction algorithms such as spanning trees, one could minimize network traffic spanning sites - why does bandwidth then matter so much?

In fact, the optimization of network latency/throughput as illustrated in the article for deep learning is no good motivation for this approach - as the communication pattern appears not to fit to what is provided by the algorithm (which tries to maximize communication performance inside a  set of nodes). 

== Scheduling Algorithm ==

The description of the Algorithm is not sufficient. (Line 398).The algorithmic description lacks clarity and must be improved. 

The network topology in the selected zone decided about expected performance. Here a monitoring system is used. Tools such as Hadoop use rack-awareness schemes avoiding to measure network. While a dynamic concept as shown in the article brings merits, static concepts should be discussed in the related work section.

According to the description in Step 4, only one zone is selected, hence, there is no multi-zone/cross-spanning deployment - which the algorithm described to be used for.

Maybe I missunderstand sth. <-> update the description and motivation in the article, please.

In the abstract, it says: "distributed work is placed in the same zone as much as possible to reduce network delay"

== Other suggestions ==

Figure 6, unfavorable empty page, rotate by 90°

285. The scheduling algorithm is not clear here: "The scheduler

selects the optimal node(s) through scoring based on this value and delivers it to the Kubernetes master server."

Add a forward reference here (described in Sec. 3.2).

Why does Eq.3 look different? 40 * v instead of 40 - ? 

Formatting of Line 349+ needs to be fixed. Hard to read.

398 => algorithm formatting is broken, use a color scheme/bold font for pseudocode constructs.

I would suggest to embed the general description as comment into the pseudocode. This simplifies relating code with the description.

Figure 10 is very blurry.

Also, it appears to be a bit too detailed here.

524 21. %

Many minor grammatical mistakes (in the beginning) of the paper - most of the time, the meaning can be inferred, though. Use grammarly to fix the most relevant ones. I report some below.

Abstract

The performance [is] often limited by heavy traffic like[s] many stochastic gradient descent transfers for distrib-

uted [communication].

 In this paper, we are focusing [on the] actual network especially [in a] Distributed HPC environment.

"if cluster nodes are assigned to different regions" => the semantics of different regions doesn't make sense here

Define zone/region here.

"The performance of distributed deep learning is improved through the proposed scheduler."

=> quantify this here

45. the capabilities [of] a

59. in the first iteration. => Sentence should continue

Figure 2: fianl => final?

Round 2

Reviewer 3 Report

Overall the paper improved. There are two issues that should be improved regarding Response 3 and Response 4.

Comments:

348: , so the value was minus when calculating the score.

Response 3: The number of comments added was rather minimal. 

Get Available Nodelist and ... Why is that line needed? It were arguments already.

R4: I don't see the description of Response 4 fitting to the algorithm.  Why is it working as described in Response 4?

Figure 11-14 are blurry.

"in Mbps" instead of  "( Mbps )"